# Multi-Grained Knowledge for Retrieval-Augmented Question Answering on Hyper-long Contexts

## Abstract

In the task of hyper-long context question answering (QA), a key challenge is extracting accurate answers from vast and dispersed information, much like finding a needle in a haystack. Existing approaches face major limitations, particularly the input-length constraints of Large Language Models (LLMs), which hinder their ability to understand hyper-long contexts. Furthermore, Retrieval-Augmented Generation (RAG) methods, which heavily rely on semantic representations, often experience semantic loss and retrieval errors when answers are spread across different parts of the text. Therefore, there is a pressing need to develop more effective strategies to optimize information extraction and reasoning. In this paper, we propose a multi-grained entity graph-based QA method that constructs an entity graph and dynamically combines both local and global contexts. Our approach captures information across three granularity levels (i.e., micro-level, feature-level, and macro-level), and incorporates iterative retrieval and reasoning mechanisms to generate accurate answers for hyper-long contexts. Specifically, we first utilize EntiGraph to extract entities, attributes, relationships, and events from hyper-long contexts, and aggregate them to generate multi-grained QA pairs. Then, we retrieve the most relevant QA pairs according to the query. Additionally, we introduce LoopAgent, an iterative retrieval mechanism that dynamically refines queries across multiple retrieval rounds, combining reasoning mechanisms to enhance the accuracy and effectiveness of answering complex questions. We evaluated our method on various datasets from LongBench and InfiniteBench, and the experimental results demonstrate the effectiveness of our approach, significantly outperforming existing methods in both the accuracy and granularity of the extracted answers. Furthermore, it has been successfully deployed in online novel-based applications, showing significant improvements in handling long-tail queries and answering detail-oriented questions.

## 1 Introduction

Long documents often contain a wealth of critical information, particularly in fields such as law, medicine, or finance. The task of hyper-long context question answering (QA) (Georgiev et al., 2024; Wang et al., 2024a) requires models to process vast amounts of information while maintaining precise contextual understanding, which is pivotal for advancing the capabilities of Large Language Models (LLMs) to process and understand extensive textual data. In recent years, LLMs, such as GPT-4 (Brown, 2020), LLaMA (Touvron et al., 2023a), (Touvron et al., 2023b), and PaLM (Chowdhery et al., 2023), have demonstrated exceptional performance in tasks involving dialogue generation and reasoning (Dagdelen et al., 2024). These models are capable of generating knowledge-rich responses and have broad applications across various domains.

Although LLMs have considerable capabilities, they still struggle to process hyper-long texts over 100K tokens (Brown et al., 2020). It is challenging for them to maintain contextual coherence and effectively capture long-range dependencies when the input text surpasses the predefined window length (Li et al., 2023a). To alleviate the problem of processing hyper-length texts, some attention mechanisms (e.g., long-range attention mechanisms) have been widely integrated into models to capture contextual information across long sequences (Peng et al., 2024). However, training

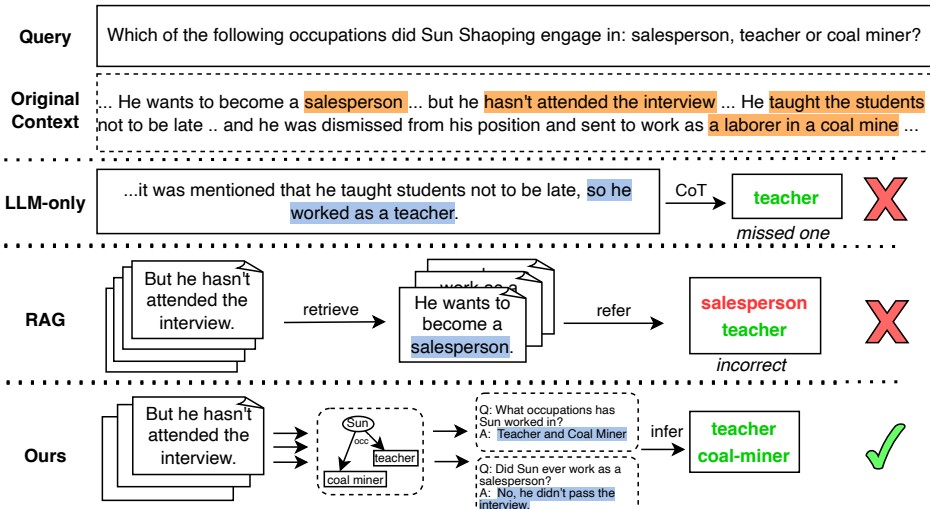

Figure 1: Comparison of QA approaches facing challenges: LLM-only is limited by context length constraints and RAG methods rely on semantic representations. Our approach integrates multi-grained knowledge with RAG, effectively overcoming these challenges to provide accurate answers.

LLMs from scratch with extended context capabilities incurs prohibitive computational costs, requiring substantial hardware resources and extensive time. To mitigate these costs, techniques such as chunking and context window expansion (Chen et al., 2023b) have been introduced to expand the context-processing capabilities of LLMs without significantly increasing computational costs. Some researchers have proposed various context extension techniques, such as long-context fine-tuning methods like LongChat (Li et al., 2023a) and Longlora (Chen et al., 2023d). These approaches adapt pre-trained models for extended context usage, thus reducing the need for training from scratch. However, the maximum context length these methods can handle remains limited, typically reaching only several hundred thousand tokens (Zhang et al., 2024).

Moreover, LLMs often exhibit high hallucination rates when applied to Knowledge-Intensive Generation tasks (Petroni et al., 2020), especially in cases where they must deal with unstructured or domain-specific datasets. These limitations are particularly evident when extracting fine-grained or deep information from large-scale texts, where the sheer volume and dispersed nature of relevant details hinder the models' ability to retrieve accurate and comprehensive answers. As shown in Figure 1, the text "work as a laborer in a coal mine" appears later in the text, falling beyond the context window and leading to a lack of understanding. Although Retrieval-Augmented Generation (RAG) (Lewis et al., 2020; Li et al., 2023c; Wang et al., 2023) methods attempt to address this by retrieving relevant information from various chunks, their effectiveness heavily relies on the precision of extracted embeddings to map queries into relevant documents. In the provided example, while the model correctly recalls that "Sun wants to become a teacher" from one chunk, it fails to integrate the information from another chunk indicating that he did not succeed in securing the position. These chunked retrieval techniques are often too coarse-grained to capture the nuanced contextual details and can result in incorrect conclusions and misinterpretations.

In this paper, we propose a **M**ulti-grained **K**nowledge **R**etrieval-**A**ugmented **G**eneration (i.e., *MKRAG*) for hyper-long context question answering. Specifically, we first extract fine-grained entities from hyper-length texts, covering entities and their corresponding attributes, relationships, and events. This extraction process goes beyond merely identifying core entities; it also captures the surrounding contextual information, ensuring that the model can synthesize multiple layers of detail when responding to complex queries. Then, we employ context aggregation algorithms to integrate both local and global contexts of the same entity, and utilize an LLM, EntiGraph, to generate multi-grained QA pairs (i.e., micro-level, feature-level, and macro-level). This multi-level strategy not only ensures that the model produces highly accurate answers for complex queries, but also mitigates the fragmentation of information that often hampers traditional methods. Finally, we propose LoopAgent, an iterative retrieval mechanism that refines queries over multiple retrieval rounds, com-

bining advanced reasoning mechanisms to enhance the retrieval and answering accuracy for complex queries. By integrating multi-grained knowledge with retrieval-augmented LLMs, our approach not only overcomes the limitations posed by traditional models' context window size but also transcends the dependency on semantic representations in retrieval-based methods by fully leveraging the rich contextual information embedded in long texts. In summary, our contributions are as follows:

1. We propose a Multi-grained Knowledge Retrieval-Augmented Generation approach, MKRAG, for hyper-long context question answering, which combines multi-grained entity graphs with iterative retrieval and reasoning. By aggregating multi-grained information and refining the retrieval process, our approach maintains higher consistency and overcomes the length limitations posed by context window constraints in traditional models.

2. We introduce LoopAgent, an iterative retrieval mechanism that refines queries over multiple retrieval rounds, which combines advanced reasoning mechanisms to enhance the retrieval and answering accuracy and addresses the information loss in traditional single-round retrieval strategies, especially in complex multi-entity scenarios.

3. We validate the effectiveness of our approach on several benchmark datasets, including Long-Bench and InfiniteBench. The experimental results demonstrate that our model consistently outperforms state-of-the-art methods in terms of accuracy and consistency for complex long-text question-answering. Compared with previous methods, our approach enhances precision in information capture while avoiding the high computational costs associated with expanding context windows or extensive fine-tuning of large models.

4. Our approach dynamically handles real-time knowledge and private data queries without relying on continuous model updates. It has been successfully deployed in various real-world applications, particularly in vertical industries that require precise handling of long-tail queries and detailed information extraction, highlighting its practical value and applicability.

## 2 RELATED WORK

The task of question answering with LLMs can be divided into two categories: long-context model optimization and RAG techniques. This section will provide a detailed overview of these two approaches and introduce how they enhance performance in long-context QA tasks.

### 2.1 LONG-CONTEXT MODEL OPTIMIZATION APPROACHES

Early approaches to long-context tasks focus on optimizing LLMs during pretraining to handle longer inputs. Many methods (Zaheer et al., 2020; Wang et al., 2020; Chen et al., 2023a; Xiao et al., 2023; Mohtashami & Jaggi, 2023; Tworkowski et al., 2024) introduce attention mechanisms and positional encoding to extend context length. For example, Press et al. (2022) proposes to bias query-key scores based on token distance, allowing models to handle longer sequences. Similarly, xPOS (Sun et al., 2023) improves long-sequence extrapolation by introducing rotational positional encoding. These methods extend model context length during pretraining to handle long input sequences. Besides, architectural innovations, like sparse patterns in transformers (Beltagy et al., 2020; Kitaev et al., 2020), are developed to reduce memory and computation demands for processing longer sequences using sparse attention mechanisms.

While pretraining methods improve long-sequence handling, they are computationally expensive, especially for sequences over 100k tokens, due to significant memory, storage, and processing demands. To mitigate this, researchers (Chen et al., 2023c; Tworkowski et al., 2024) focus on positional encoding optimization and fine-tuning strategies instead of full retraining. RoPE (Su et al., 2024) adjusts positional encodings during inference, LongLoRA (Chen et al., 2023d) combines low-rank adaptation with attention optimization, and LongChat (Li et al., 2023b) fine-tunes using rotary embeddings and conversational datasets. However, these methods still face challenges in maintaining coherence and understanding for extremely long sequences.

## 2.2 RETRIEVAL-AUGMENTED APPROACHES

RAG efficiently integrates external knowledge retrieval with LLM, making it well-suited for unstructured or domain-specific datasets. This process involves retrieving pertinent information from a vast data corpus in response to a user query, which is then fed into the LLM to enrich the generation process. The RAG strategy typically utilizes chunk-based retrieval (Guu et al., 2020; Lewis et al., 2020; Borgeaud et al., 2022; Izacard & Grave, 2021; Ram et al., 2023; Finardi et al., 2024; Setty et al., 2024), dividing long texts into smaller segments and summarizing them to improve indexing accuracy. This kind of approach relies on precise embeddings and retrieval techniques. For example, (Lewis et al., 2020) introduces dense vector representations to better align queries with document segments. To address limitations in retrieval fusion, FiD (Izacard & Grave, 2021) integrates retrieved passages during the generation process to synthesize more accurate responses.

However, chunk-based methods have limitations, such as losing coherence when dividing text and relying heavily on the semantic understanding of both the query and document. They may also retrieve fragmented or incomplete information, resulting in fragmented or incomplete answers. To overcome these limitations, graph-based RAG methods (Pan et al., 2024; Wang et al., 2024b; Zhang et al., 2023; Sen et al., 2023; Xu et al., 2024; Jiang et al., 2024; Shao et al., 2023; Hu et al., 2024; Ma et al., 2024) incorporate relational information from knowledge graphs, improving reasoning across chunks and enhancing information integration. For example, GNN-RAG (Mavromatis & Karypis, 2024) enhances retrieval by preserving relationships between entities using structured graph information. Similarly, (Ma et al., 2024) introduced a RAG framework guided by knowledge graphs, leveraging multi-hop relationships and key entities to address long-range dependencies and ensure logical consistency in complex reasoning tasks. Graph-based RAG methods advance reasoning but struggle with simplistic graphs and shallow entity relationships, limiting complex reasoning and context-dependent tasks. These limitations lead to knowledge loss and slower inference. Our work introduces multi-grained knowledge with RAG to address these challenges.

## 3 TASK DEFINITION

**Definition: Knowledge-Intensive QA for Hyper-Long Contexts.** Hyper-long contexts refer to contexts whose length is over 100K tokens, making it infeasible to process the entire context. The task aims to generate an accurate answer $A$ to a query $Q$ based on such extensive contexts, where the pertinent information is often dispersed across multiple sections. To address the complexities posed by this task, we propose a novel multi-grained entity graph-based QA generation framework. This method decomposes the hyper-long document $T$ into a set of sub-blocks $T_i$, from which entities, attributes, relationships, and event-related information are extracted to construct an entity set $E$. By dynamically aggregating this information across different levels of granularity, our approach facilitates the generation of precise QA pairs. This progress can be formulated as following:

$$A = f(T, Q; \Theta) = \arg\max P(A \mid Q, E_{\text{micro}}, E_{\text{feature}}, E_{\text{macro}}; \Theta) \tag{1}$$

where $f(\cdot)$ represents our multi-grained entity graph-based QA generation model, and $\Theta$ represents the model parameters. QA pairs are constructed on three granularities of entity information to store local and global information: micro-granularity ($E_{\text{micro}}$), feature-level granularity ($E_{\text{feature}}$), and macro-granularity ($E_{\text{macro}}$). By synthesizing information from these levels, the model effectively reduces information loss and mitigates semantic inconsistency, thus enhancing the accuracy of QA generation in hyper-length contexts.

## 4 METHOD

As illustrated in Figure 2, we propose a comprehensive approach for knowledge extraction from hyper-long contexts and response generation. The framework consists of two core components: multi-grained knowledge generation and iterative retrieval agent. In the first component, the model targets the extraction of entities, attributes, relationships, and events from extended contexts, generating multi-grained knowledge. This involves constructing micro, feature, and macro-level QA pairs, which are essential for capturing intricate details within complex, lengthy texts. Sections 4.1 through 4.4 provide a detailed explanation of this process. The retrieval and agent components are covered in Section 4.5.

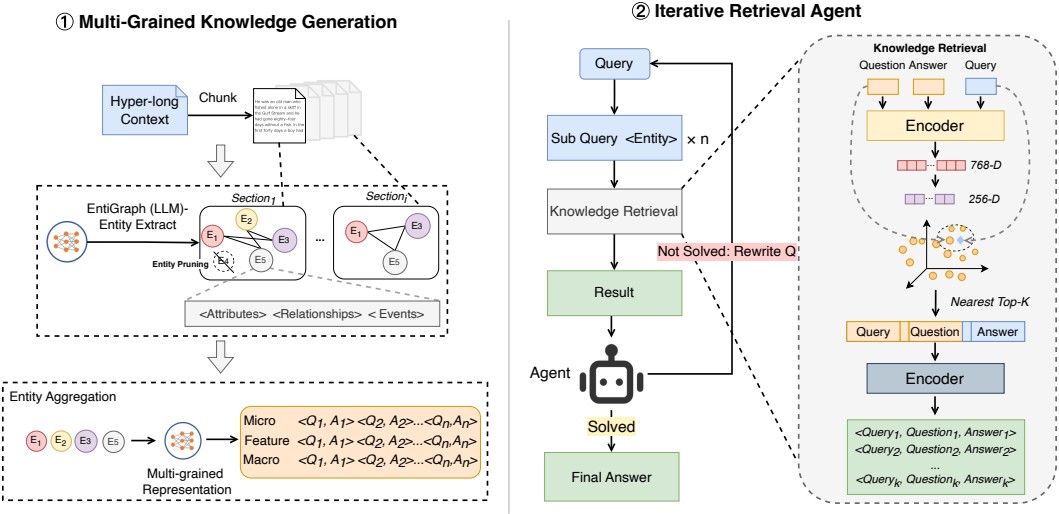

Figure 2: The method has two main components: multi-grained knowledge generation and iterative retrieval agent, where hyper-long context is processed to capture entities with attributes, relationships and events, and an iterative retrieval agent that refines queries to retrieve relevant information for generating the final answer.

## 4.1 ENTITY EXTRACTION

Given a long text $T$ consisting of multiple sentences, we group these sentences into chunks, where each chunk $T_i$ is composed of a consecutive sequence of sentences. Formally, the segmentation process divides the text into $m$ sub-blocks, expressed as:

$$T = \{T_1, T_2, ..., T_m\}, \quad \text{where} \quad T_i = \{s_{l_{(i-1)}+1}, s_{l_{(i-1)}+2}, ..., s_{l_i}\} \tag{2}$$

Here, $l_i$ represents the index of the last sentence in each chunk $T_i$, allowing the chunk sizes to vary as needed. For each chunk $T_i$, we employ an LLM, named EntiGraph, to extract entities along with their associated attributes, relationships, and events. The entity extraction process can be implemented in the following two ways:

- **Few-shot Learning Adaptation.** We incorporate both positive and negative samples into the prompt design, which can enhance the model's ability to accurately extract entities across different domains. Positive samples represent the entities the model should extract, while negative samples help reduce the likelihood of incorrect or irrelevant extractions.

  For each sub-block $T_i$, we utilize a LLM to perform entity extraction. It generates an entity set $E_i$ by leveraging the few-shot learning approach, where the positive and negative samples are integrated into the prompts used for inference. The extraction process is formalized as:

$$E_i = \arg\max_E P(E \mid T_i, \mathrm{S}; \theta_{\mathrm{pre}}) \tag{3}$$

  where $\theta_{\mathrm{pre}}$ represents the model parameters, and $E_i$ denotes the set of entities extracted from sub-block $T_i$. Samples $S$ refers to the positive and negative samples included in the prompt. By incorporating these samples, the model better adapts to varying domain-specific contexts, reducing extraction errors and improving accuracy in entity recognition tasks across diverse domains.

- **Specialized Model Fine-tuning.** Another approach is to fine-tune the model on domain-specific datasets to enhance its performance in entity extraction tasks. The core of this method lies in constructing diverse training samples that comprehensively cover entity types, complex relationships, and fine-grained attributes relevant to the target domain. These training samples not only include common entity categories but also encompass similar entities in different contexts, complex entity relationship structures, and nested entities with multiple attributes. This diverse sample-construction significantly improves the model's ability to understand and extract fine-grained semantic information. Specifically, the entity extraction task can be formulated as: $E_i = \mathrm{EntiGraph}(T_i; \theta_{\mathrm{ft}})$, where $\theta_{\mathrm{ft}}$ denotes the fine-tuned model parameters.

Furthermore, to better utilize the extracted entity information, the fine-tuning process can incorporate training samples that map entity information to QA pairs. These samples align the extracted entities and their attributes with corresponding QA pairs, enhancing the model's understanding of contextual information. In Section 4.4, this fine-tuning strategy helps the generation of more accurate and coherent QA outputs within the multi-grained representation framework. It is worth noting that this fine-tuning strategy enables the use of a compact model with a parameter size of around 1B to achieve excellent performance.

## 4.2 Entity Aggregation and Temporal Contextualization

To effectively capture the temporal relationship between entities in a long document, we propose to aggregate entities based on their occurrences across different sub-blocks while preserving their temporal order. An entity $e_j$ may appear in multiple sub-blocks $T_i$, with each occurrence potentially associated with different attributes, relationships, and events. To build a comprehensive understanding of the entity's role throughout the document, these occurrences are aggregated chronologically, ensuring that both the entity's evolution and temporal context are maintained.

The aggregation process collects all instances of an entity across the sub-blocks and organizes them according to their order of appearance in the text. For each entity $e_j$, the attributes, relationships, and events from different sub-blocks are combined, preserving the associated temporal information to form a unified, time-sensitive representation. The aggregated entity is formalized as:

$$E' = \bigcup_{j=1}^{N} \{(e_j, \{a_j, r_j, ev_j, t_j\})\} \tag{4}$$

where $a_j$, $r_j$, and $ev_j$ denote the sets of attributes, relationships, and events associated with $e_j$, respectively. The $t_j$ refers to the timestamp indicating when the entity appears in the text. This aggregation method ensures that temporal contextual information is retained during the entity aggregation, preventing information loss or inconsistency.

## 4.3 Entity Pruning

To enhance computational efficiency and reduce redundancy, we developed an entity pruning algorithm aimed at improving the precision of QA pair generation by eliminating ambiguous or superfluous entities. We define a pruning function $P(e_j)$ to determine whether an entity should be retained, and if the importance of the entity $e_j$ falls below a predefined threshold $\tau$, the entity is pruned:

$$P(e_j) = \begin{cases} 1, & \text{if } \sum_{k=1}^{K} w_k \cdot I(e_j^k) \geq \tau \text{ and } t_{ij} \text{ is a specific timestamp} \\ 0, & \text{otherwise} \end{cases} \tag{5}$$

where $w_k$ represents the importance weight of the entity, assigned by a large language model (Ernie-3.5-8k) based on predefined scoring rules to evaluate the entity's significance (details provided in Appendix A.3.2); and $I(e_j^k)$ is an indicator function that signifies whether the entity attribute $e_j^k$ exists. Additionally, temporal expressions associated with $e_j$ are checked: if $t_{ij}$ is a generalized temporal expression, the entity is either pruned or the temporal information is discarded, unless a clear and specific time point is provided (e.g., "1981 AD").

The selection of the pruning threshold ($\tau$) was determined based on a detailed analysis of its impact on entity pruning effectiveness. As shown in Table 7 in the Appendix, we conducted experiments to evaluate the trade-offs between the number of retained entities and performance metrics (Precision, Recall, and F1). This analysis demonstrates that a threshold of $\tau = 0.5$ achieves the best balance between Precision and Recall.

## 4.4 Multi-grained Representation

The entities and information are modeled as nodes in a graph, and generating QA pairs corresponds to finding the shortest paths between these nodes. Based on aggregated entity information, QA pairs are constructed at three different granularities to capture semantic-rich information. The QA generation process can be unified into a single formulation:

$$(Q, A) = \arg\max_{q,a} P(q, a \mid e, \Gamma) \tag{6}$$

where $\Gamma$ represents the information set at a particular granularity level:

- **Micro-Grained Granularity.** $\Gamma = \kappa$, with $\kappa \in \{\alpha, \rho, \epsilon\}$ representing a specific attribute ($\alpha$), relationship ($\rho$), or event ($\epsilon$).
- **Feature-Level Granularity.** $\Gamma = \mathcal{K}$, where $\mathcal{K} \in \{\mathcal{A}, \mathcal{R}, \mathcal{EV}\}$ denotes the complete set of attributes, relationships, or events within a given dimension.
- **Macro-Grained Granularity.** $\Gamma = G = \{\mathcal{A}, \mathcal{R}, \mathcal{EV}\}$, representing the global information set of the entity.

This unified formulation captures QA generation across all granularities by varying the information set $\Gamma$. The multi-grained QA generation framework dynamically adjusts the granularity of QA pairs based on the amount of information available for each entity, ensuring both efficiency and accuracy. For entities with sparse information (e.g., niche or less significant entities), only macro-level QA pairs are generated to reduce computational overhead.

Let the information set for an entity $e$ be $I(e) = \{\mathcal{A}, \mathcal{R}, \mathcal{EV}\}$, with size $|I(e)|$. The granularity of QA pairs is determined by $|I(e)|$ as follows:

$$(Q, A) = \begin{cases} (Q^{\text{macro}}, A^{\text{macro}}), & \text{if } |I(e)| < \tau_1 \\ (Q^{\text{macro}}, A^{\text{macro}}), (Q^{\text{feature}}, A^{\text{feature}}), & \text{if } \tau_1 \leq |I(e)| < \tau_2 \\ (Q^{\text{macro}}, A^{\text{macro}}), (Q^{\text{feature}}, A^{\text{feature}}), (Q^{\text{micro}}, A^{\text{micro}}), & \text{if } |I(e)| \geq \tau_2 \end{cases} \quad (7)$$

where $\tau_1$ and $\tau_2$ are thresholds defining the granularity levels. This adaptive strategy optimizes computational efficiency by tailoring the QA generation process to the richness of the entity's information. The distribution of representation granularity across datasets is provided in Appendix Figure 3, highlighting the proportions of basic, feature, and global representations.

### 4.5 ITERATIVE RETRIEVAL AGENT

The LoopAgent uses a multi-round iterative retrieval strategy, dynamically adjusting queries to refine results with each round. This approach overcomes the limitations of single-round retrieval, which often misses critical information in complex, multi-entity scenarios.

The retrieval process begins by decomposing the original query $Q$ into $K$ sub-queries, each derived from a specific entity in the query. These sub-queries are processed in two phases: Retrieval and Re-ranking. In the Retrieval stage, the model employs a 12-layer encoder to process the query, and the question and answer are concatenated and fed into another 6-layer encoder. By encoding the query and QA pairs separately, the dual-encoder model (Yates et al., 2021; Fan et al., 2022; Luan et al., 2021) calculates their similarity in the shared vector space, enabling the selection of the most relevant QA pairs that align with the query intent. The system retrieves and ranks the top-$K_1$ relevant question-answer pairs from a corpus based on their similarity to the sub-query:

$$\text{TOP}_K = \text{argmax}_{i=1\ldots n} \text{sim}(f(\mathbf{Q}; \alpha), g(\mathbf{QA}_i; \beta)) \quad (8)$$

where $\mathbf{Q}$ is the input query, $\mathbf{QA}_i$ is the candidate from the corpus, and $f$ and $g$ are the encoders parameterized by $\alpha$ and $\beta$. The similarity function sim measures the relevance between the encoded query and candidate pairs. After retrieving the top-$K_1$ results, a Re-ranking stage refines them using a cross-encoder (Qiao et al., 2019), recalculating relevance and producing a new top-$K_2$ ranking by incorporating additional attributes.

If the first-round results lack sufficient information, the agent identifies gaps and generates an adjusted query $RewriteQ$. A second retrieval round based on $RewriteQ$ produces a new result set $R_2$, which is merged with $R_1$ to create a more comprehensive set. Through multiple rounds of retrieval and query adjustment, LoopAgent captures as much relevant information as possible. Once sufficient data is gathered, the results are passed through a language model to generate a fluent, accurate, and coherent final answer, balancing efficiency with precision.

## 5 EXPERIMENTS

In this section, we present the details of experimental setup, with the results and a detailed analysis.

Table 1: Comparison with state-of-the-art on InfiniteBench dataset.

| Methods | GPT-4 | YaRN-Mistral | Kimi-Chat | Claude 2 | Yi-6B-200K | Yi-34B-200K | ChatGLM3-128K | **Ours** |
|---|---|---|---|---|---|---|---|---|
| F1 | 25.96% | 16.98% | 17.93% | 9.64% | 15.07% | 13.61% | <5% | **45.61%(↑)** |

Table 2: Comparison with state-of-the-art on MultiFieldQA and DuReader datasets.

| Methods | GPT-3.5 Turbo-16k | Llama2-7B chat-4k | LongChat-v1.5 7B-32k | XGen 7B-8k | InternLM 7B-8k | ChatGLM2 6B-32k | Vicuna-v1.5 7B-16k | ChatGLM3 6B-32k | **Ours** |
|---|---|---|---|---|---|---|---|---|---|
| MultiFieldQA-en(F1) | 52.3 | 36.8 | 41.4 | 37.7 | 23.4 | 46.2 | 38.5 | 51.7 | **63.3(↑)** |
| MultiFieldQA-zh(F1) | 61.2 | 11.9 | 29.1 | 14.8 | 33.6 | 51.6 | 43.0 | 62.3 | **65.6(↑)** |
| DuReader(Rouge-L) | 28.7 | 5.2 | 19.5 | 11.0 | 11.1 | 37.6 | 19.3 | **44.7** | 31.4 |

## 5.1 EXPERIMENTAL SETUP

In this study, we utilize two benchmark evaluations: LongBench (Bai et al., 2023) and InfiniteBench (Zhang et al., 2024), each comprising distinct datasets aimed at assessing language models' performance in long context understanding. Detailed descriptions of these datasets are provided in Appendix A.1. Based on the official evaluation metrics, we assess the model's performance on each task as follows: The DuReader (He et al., 2018) was evaluate using ROUGE scores (ROUGE-1/2/L) as the primary metrics. For the MultiFieldQA-zh, MultiFieldQA-en, and tasks within InfiniteBench, F1 scores are used to evaluate the model's accuracy in question-answering. The baseline model used in our study is Ernie-3.5-8k, with a context token limit of 4k.

## 5.2 COMPARISONS WITH STATE-OF-THE-ARTS

In this study, we evaluate the performance of multiple models on the InfiniteBench and LongBench benchmarks, with a particular emphasis on long-context comprehension and multi-domain question answering. The results indicate that our model's strengths in long-context understanding become increasingly evident as text length grows. A comprehensive analysis is presented below.

**Performance on Long Context Datasets.** As shown in Table 1, our model consistently outperforms baseline models on the hyper-long context dataset, specifically InfiniteBench (Zh.QA). It achieves an F1 score of 45.61%, surpassing GPT-4 (OpenAI, 2023) (25.96%) and other models such as Kimi-Chat (AI, 2023) (17.93%) and Claude 2 (Anthropic, 2023) (9.64%). These results highlight our model's ability to maintain relevance as the input text length exceeds 100K tokens, particularly in tasks requiring the processing of hyper-long contexts. This demonstrates the model's superior capability in understanding and reasoning over extremely long text.

**Performance on Shorter Text Datasets.**

For datasets with shorter texts like MultiFieldQA-en, MultiFieldQA-zh, and DuReader, where document lengths fit within the context window of baseline models (e.g., GPT-3.5-Turbo-16k and ChatGLM3-6B-32k (Zeng et al., 2022)), our model remains competitive (Table 2). In MultiFieldQA, it achieves the highest F1 scores in English (63.3%) and Chinese (65.6%), showcasing strong generalization and robust multilingual comprehension.

In DuReader, our model attains a ROUGE-L score of 31.4, below ChatGLM3-6B-32k (44.78%) but still demonstrates strong generative abilities for long Chinese documents. However, ROUGE-L, as a lexical metric, may miss semantic accuracy. For instance, given the query "What rank is Gatanothor in the monster list?" with the reference answer "Top 7. Gatanothor (Ruler of Darkness)," our model's response, "Ranked seventh in the monster list," is semantically correct despite a ROUGE-L score of 0, highlighting ROUGE-L's limitations. To address this, we propose an LLM-based evaluation (Section 5.3) to assess semantic correctness, offering a more comprehensive performance measure.

## 5.3 ABLATION STUDY

We conducted ablation experiments to evaluate the contributions of different components in the proposed model, comparing the baseline LLM (Ernie-3.5-8k), chunk-based retrieval combined with the baseline, the EntiGraph module, and the Multi-Grained Representation Module, culminating in

Table 3: Comparison of Different Methods across Datasets

| Methods | InfiniteBench(Zh.QA) | MultiFieldQA-en | MultiFieldQA-zh | DuReader |
|---|---|---|---|---|
| Baseline (Ernie-3.5-8k) | <5% | 27% | 56% | <5% |
| RAG | <5% | 16% | 35% | 71% |
| **Ours** | **53%(↑)** | **79%(↑)** | **78%(↑)** | **73%(↑)** |

Table 4: Evaluation of Entity Extraction Methods.

| Methods | Accuracy (%) | Recall (%) | F1 Score (%) | Extracted Entities (#) |
|---|---|---|---|---|
| DeepKE | 52.9 | 18 | 27.4 | 164 |
| Ours | 97.4 | 82 | 89.3 | 1,018 |

the full MKRAG framework. Table 3 summarizes performance across four datasets: InfiniteBench, MultiFieldQA-en, MultiFieldQA-zh, and DuReader.

The baseline LLM (Ernie-3.5-8k), constrained by its 4k token limit, showed limited performance on long-context datasets, achieving only 27% accuracy in MultiFieldQA-en and less than 5% in DuReader due to token overflow. Adding chunk-based retrieval (500 tokens per chunk) improved accuracy, particularly in DuReader (71%), benefiting from smaller dataset sizes and distinct features. However, this approach struggled with tasks requiring comprehensive contextual understanding, as it heavily relied on retrieving similar chunks without effectively integrating dispersed information.

**EntiGraph.** To evaluate entity extraction reliability, we compared our method with DeepKE (Zhang et al., 2022) using GPT-4-extracted ground truth (518 entities). Metrics included accuracy, recall, F1 score, and total entities extracted. As shown in Table 4, our method outperformed DeepKE across all metrics, achieving 97.4% accuracy, 82% recall, and 89.3% F1, compared to DeepKE's 52.9%, 18%, and 27.4%, respectively. Our approach also extracted more entities (1,018 vs. 164), demonstrating higher precision and recall, effectively addressing sparse or ambiguous entity relationships. This improvement is critical for downstream tasks like knowledge graph construction and question answering, ensuring robust performance.

**Multi-grained Representation Module.** To further evaluate the impact of Multi-Grained Representation, we conducted additional ablation experiments focusing on micro, feature, and macro granularities across datasets. Table 5 illustrates the improvements in accuracy and F1 score at each granularity, demonstrating the significant contributions of MKRAG. For instance, in the MultiFieldQA-zh dataset, macro-level accuracy and F1 score increased by 16% and 13.3%, respectively, showcasing the model's ability to capture global contextual information. Similarly, micro-level improvements highlight enhanced detailed reasoning, particularly in the InfiniteBench dataset. These results reaffirm the necessity of multi-grained representation for handling hyper-long contexts.

**MKRAG.** Our full model, MKRAG, excels in all datasets, particularly in long-context tasks. In InfiniteBench, it achieves 53% accuracy, surpassing baselines and chunk retrieval, with the EntiGraph module enhancing representation for hyper-long inputs. For shorter datasets like MultiFieldQA-zh and MultiFieldQA-en, despite less pronounced gains, MKRAG achieves 78% and 79% accuracy, nearly doubling the baseline in the latter. Ablation studies confirm its strength in long-context tasks, overcoming token limits and query-document mapping issues in chunk retrieval. The multi-grained RAG approach proves effective for high accuracy across diverse datasets.

## 5.4 EFFICIENCY EVALUATION

To evaluate the computational efficiency of our proposed iterative retrieval agent (Section 4.5), we conducted experiments on the InfiniteBench (a long-document dataset) and MultiFieldQA-zh (a relatively short-document dataset). Table 6 reports inference time, accuracy, F1 score, and average iteration rounds under a maximum of two retrieval iterations. Our method achieves improvements in accuracy and F1 scores compared to GPT-3.5 Turbo-16k and RAG (Ernie-3.5-8k), while maintaining competitive inference times across both datasets. The average iteration rounds demonstrate

Table 5: Performance of Multi-Grained Representation on Various Datasets

| Dataset | Granularity | Acc (%) | F1 (%) | MKRAG (Acc, %) | MKRAG (F1, %) | ΔAcc (%) | ΔF1 (%) |
|---|---|---|---|---|---|---|---|
| **DuReader** | micro | 63 | 22.4 | | | +10 | +9.0 |
| | feature | 51 | 21.8 | 73 | 31.4 | +22 | +9.6 |
| | macro | 64 | 23.7 | | | +9 | +7.7 |
| **MultiFieldQA-en** | micro | 63 | 35.5 | | | +16 | +27.8 |
| | feature | 65 | 33.0 | 79 | 63.3 | +14 | +30.3 |
| | macro | 73 | 34.8 | | | +6 | +28.5 |
| **MultiFieldQA-zh** | micro | 63 | 47.5 | | | +12 | +18.1 |
| | feature | 59 | 44.6 | 78 | 65.6 | +16 | +21.0 |
| | macro | 74 | 52.3 | | | +1 | +13.3 |
| **InfiniteBench** | micro | 46.88 | 40.00 | | | +6.12 | +5.61 |
| | feature | 42 | 34.70 | 53 | 45.61 | +11 | +10.91 |
| | macro | 37 | 32.00 | | | +16 | +13.61 |

Table 6: Comparison of Inference Efficiency and Iterative Performance.

| Dataset | Methods | Time(s) | Acc. (%) | F1 Score (%) | Avg. Iters |
|---|---|---|---|---|---|
| **InfiniteBench (Long)** | GPT-3.5 Turbo-16k | 2.57 | 12 | 6.2 | 1 |
| | RAG (Ernie-3.5-8k) | 1.32 | 48 | 34.3 | 1 |
| | Ours | 2.03 | 53 | 45.6 | 1.34 |
| **MultiFieldQA-zh (Short)** | GPT-3.5 Turbo-16k | 1.91 | 74 | 61.2 | 1 |
| | RAG (Ernie-3.5-8k) | 1.80 | 75 | 60.7 | 1 |
| | Ours | 1.89 | 78 | 65.6 | 1.09 |

the adaptability of our iterative retrieval mechanism, dynamically refining retrieval results based on query complexity. This configuration effectively balances retrieval quality and computational efficiency, showcasing the practicality of our approach for both long and short-text scenarios.

## 5.5 ONLINE TEST

Our method underwent multiple rounds of experiments, demonstrating high accuracy in specific task domains and successfully being deployed in real-world systems. It shows significant advantages in managing updates and scaling large datasets, such as financial data and literary texts. Compared to existing large models, accuracy improves from a baseline of 33.33% to 54.91%, with detailed answer accuracy rising further to 88.24%. The F1 score increases from 15.38% to 44.72%, and user satisfaction reaches 82.35%. These results highlight the method's efficiency and accuracy in processing complex long-context and real-time information, underscoring its practical value in handling large-scale and complex data tasks.

## 6 CONCLUSION

In this study, we present a multi-grained knowledge approach for question answering over hyper-long contexts. Our method constructs a knowledge graph organizing information at micro, feature, and macro levels, enhancing LLMs' ability to process extensive, distributed data. Unlike traditional models struggling with fragmented context, it integrates fine-grained entities and context aggregation to deliver precise, rich responses. Experimental results show a 54.91% accuracy improvement in extracting dispersed details in domains like finance and literature. With plug-and-play functionality, lower costs, and real-world efficacy, our approach reduces reliance on high-parameter models while excelling in knowledge-intensive tasks.

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

# A APPENDIX

## A.1 DATASET DETAILS

**LongBench** LongBench is a multi-task, bilingual benchmark for long-context comprehension in *Chinese* and *English*, featuring tasks like single-document and multi-document QA, with task lengths ranging from 5k to 15k tokens. MultiFieldQA (en  zh) spans domains such as *legal documents*, *government reports*, *encyclopedias*, and *academic papers* in both languages. DuReader, based on Baidu Search and Zhidao queries, focuses on multi-step reasoning and generate answers in complex Chinese long-text documents, with an average length of 15,768 characters.

**InfiniteBench** InfiniteBench is a benchmark designed for hyper-long contexts (100k+ tokens), extending context length far beyond conventional tasks to challenge model capabilities in such scenarios. Zh.QA, the **longest** dataset in InfiniteBench, is based on newly curated *books*, with an average input length of **2,068.6k** tokens and an average output of 6.3 tokens.

## A.2 ADDITIONAL RESULTS

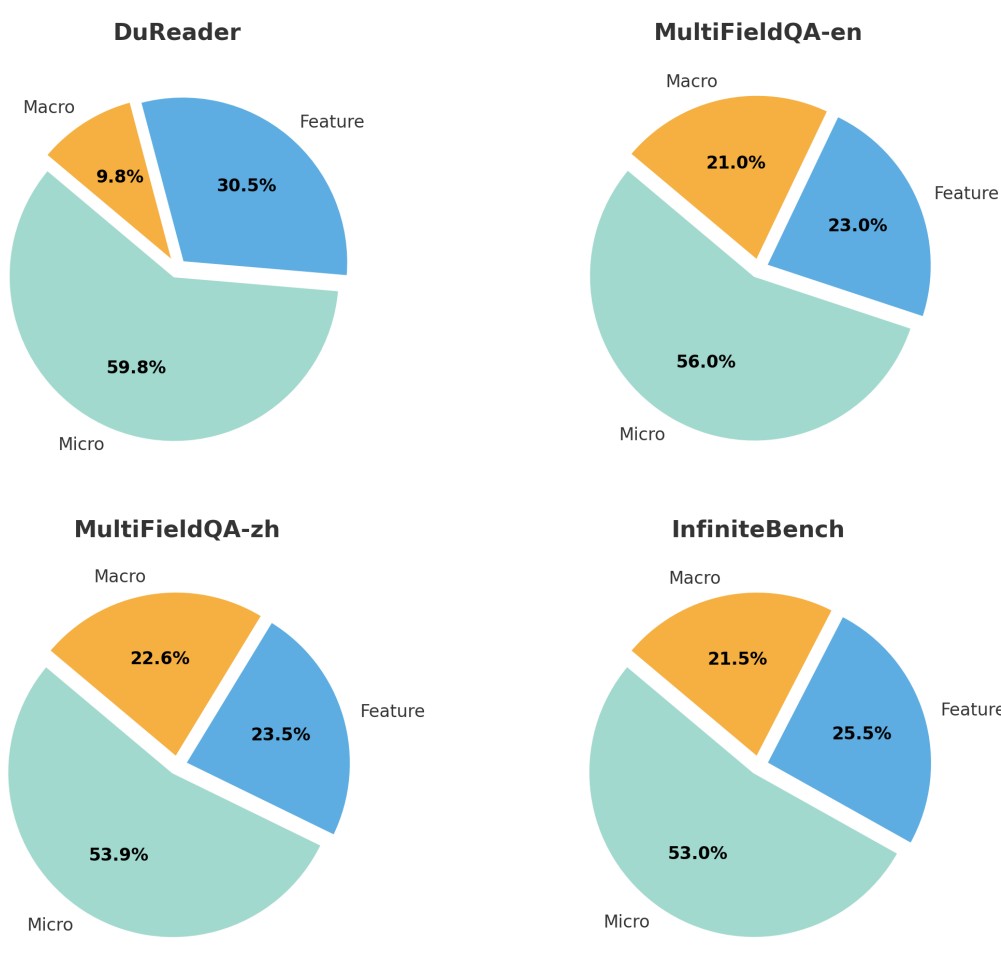

Figure 3: Distribution of Representation Granularity Across Datasets.

### A.2.1 DISTRIBUTION OF REPRESENTATION GRANULARITY

Figure 3 demonstrates the representation granularity distribution across four datasets—DuReader, MultiFieldQA-en, MultiFieldQA-zh, and InfiniteBench. The dominance of basic representations

(53.0–59.8%) across datasets reflects the inherent need for core entity information in understanding hyper-long contexts. However, the significant proportions of feature-level (23.0–30.5%) and global representations (9.8–22.6%) underline the importance of capturing nuanced and contextual information. These findings validate the necessity of multi-grained representations to address the diversity of queries and context complexities in hyper-long documents.

## A.3 ADDITIONAL EXPERIMENTAL SETUP

### A.3.1 ENTITY PRUNING THRESHOLDS

To determine the optimal threshold ($\tau$) for entity pruning, we conducted experiments to evaluate the trade-offs between Precision, Recall, and F1 score across different values of $\tau$. Table 7 presents the results, demonstrating that increasing $\tau$ reduces the number of retained entities but improves Precision by eliminating less relevant entities. While higher thresholds lead to a drop in Recall, the F1 score peaks at $\tau = 0.5$, indicating the best balance between Precision and Recall for downstream tasks. These results validate our choice of $\tau$ and its alignment with the module's objective of efficient and accurate entity selection.

Table 7: Impact of Entity Pruning Thresholds ($\tau$) on Performance Metrics

| Thresholds ($\tau$) | Precision (%) | Recall (%) | F1 (%) | nums (Entities Retained) |
|:---:|:---:|:---:|:---:|:---:|
| 0 | 86.3 | 84.6 | 85.5 | 1272 |
| 0.3 | 90.5 | 80.8 | 85.3 | 1107 |
| **0.5** | **95.2** | **80.8** | **87.5** | **688** |
| 0.7 | 88.2 | 65.4 | 75.1 | 398 |
| 1 | 90.1 | 42.3 | 57.7 | 268 |

### A.3.2 DEFINITION OF IMPORTANCE WEIGHT $w_k$

The importance weight $w_k$ in Equation (6) is assigned to each entity based on its contextual relevance and semantic characteristics, as determined by a large language model (e.g., Ernie-3.5-8k). The scoring rules are as follows:

Weight 0.3: Generic or vague entities, such as "villager" or "merchant." These entities are typically less informative and have minimal contextual contribution. Weight 0.5: Entities that are contextually relevant and describe specific subjects, such as "Zhu Bajie's wife." These entities provide clear and actionable information in the context.

Weight 0.7: Rare or specific entities with unique names or backgrounds, such as "White Bone Demon." These entities are often critical for understanding specific events or descriptions.

Weight 1.0: Core entities, which are indispensable to the context, such as "Sun Wukong." These entities are essential for reasoning and often central to the context.

### A.3.3 MODEL PARAMETERS

The configurations of three models: ErnieBot, retrieval-Ernie, and rerank-Ernie. ErnieBot's architecture and key parameters remain unpublished, with a version of 3.5. Both retrieval-Ernie and rerank-Ernie use the Transformer architecture, version 2.0. Retrieval-Ernie has 12 layers, a hidden size of 768, 12 attention heads, and employs the infoNCE loss function. Rerank-Ernie, with 6 layers, shares the same hidden size and attention heads but uses the hinge loss function, optimizing for ranking tasks.

## A.4 PROMPTS

### A.4.1 ENTITY EXTRACTION

Table 8: Model Parameters in Experiments

| Model | Architecture | Version | Layers | Hidden Size | Attention heads | Loss Function |
|---|---|---|---|---|---|---|
| Ernie-3.5-8k | Transformer | 3.5 | - | - | - | - |
| retrieval-Ernie | Transformer | 2.0 | 12 | 768 | 12 | infoNCE |
| rerank-Ernie | Transformer | 2.0 | 6 | 768 | 12 | hinge loss |

```
Please extract all specific information about the all
    ↪ entities from the following text in the style of
    ↪ in-depth reading comprehension, including
    ↪ attributes (such as the functions or
    ↪ characteristics of objects, or detailed
    ↪ descriptions of people, places, etc.),
    ↪ relationships (the logical connections between
    ↪ entities, such as cause-effect, belonging,
    ↪ contrast, etc.), and events (summarize the core
    ↪ events in which the entities participated and
    ↪ their key details).

[Guidelines]
{guidelines}

[Example]
{example}

Please analyze the following text in accordance with
    ↪ the above requirements and extract relevant
    ↪ information. If no relevant information is found,
    ↪ return {{}}:
"
{content}
"
```

### A.4.2   MULTI-GRAINED QA PAIRS GENERATION

```
{instruction}
The following are question-and-answer pairs generated
    ↪ based on provided entities and
    ↪ attributes/relationships/events.

[Guidelines]
{guidelines}

{few_shot_examples}
**Entity Relationships:** {content}

**Question:**
{question}
**Answer:**

Begin!
Entity Relationships:
"
{content}
"
```

### A.4.3   QUERY STAGE 1: DECOMPOSE SUB-QUERIES

```
Decompose the original query into multiple short
    ↪ queries based on the entity information. The
    ↪ decomposed queries must remain consistent with
    ↪ the original query and should not be expanded.
{instruction}

[Guidelines]
{guidelines}

[Example]
{examples}

[Thought]
{thought}

Begin!
Question:
"
{query}
"
```

### A.4.4  QUERY STAGE 2: LOOPAGENT

```
[Task]
Understand the Question, extract or deduce the answer
    ↪ based on the Knowledge, and if the answer cannot
    ↪ be derived, reconstruct the Question based on the
    ↪ missing information. Finally, output the answer.
Knowledge: {Knowledge}
Question: {Question}

[Observation]
{Observation}

[Example]
{examples}

Begin!
Question:
"
{Question}
"
Knowledge:
"
{Knowledge}
"
```

## A.5  EXAMPLE

### A.5.1  ENTITY EXAMPLE

```
{
    "Attributes": {
        "Personality Traits": [
            "Generous and bold",
            "Witty and humorous",
```

```
972                      "Skilled in martial arts",
973                      "Sinister but with underlying grievances"
974                  ],
975                  "Associates": "Beautiful woman (wife)",
976                  "Identity": [
977                      "Leader of the Northern Branch of Tianlong
                              ↪ Sect",
978                      "Person who met Hu Yitong years ago at
979                          ↪ Shangjiabao",
980                      "Acquainted with Miao Qiaowei and Hu
981                          ↪ Cuishan",
982                      "Head of a major martial arts sect",
983                      "Leader of Tianlong Sect",
984                      "Figure in Hua Quan Sect"
985                  ],
                  "Attire": "Luxurious",
986                  "Complexion": "Pale like paper",
987                  "Relationship": "Husband of the beautiful
                          ↪ woman",
988                  "Appearance": [
989                      "Handsome and dashing",
990                      "Long eyebrows and bright eyes, exuding
991                          ↪ elegance"
992                  ],
993                  "Character": [
                      "Charming",
994                      "Appears superior but is actually cautious",
995                      "Strategic and prudent",
996                      "Fond of teasing others"
997                  ],
998                  "Martial Arts": [
                      "Not particularly skilled"
999                  ],
1000                 "Weapons": [
1001                     "Long sword",
1002                     "Short knife",
1003                     "Treasure sword",
1004                     "Long sword and Tianlong Treasure Blade"
                  ],
1005                 "Behavioral Traits": [
1006                     "Suddenly standing up",
1007                     "Gripping the hilt of a long sword at the
                              ↪ waist, drawing it five inches with a
1008                          ↪ clang, and returning it to the
1009                          ↪ scabbard",
1010                     "Saying softly, 'Lanmei, l e t s  go.'",
1011                     "Eyes fixed on the silver scabbards in the
                              ↪ carriage",
1012                     "Speaking with a trembling voice"
1013                 ],
1014                 "Character Traits": [
1015                     "Dashing and efficient",
1016                     "Fearful inside",
1017                     "Greedy for silver scabbards"
                  ],
1018                 "Characteristics": [
1019                     "Extremely sinister schemes",
1020                     "Terrified of the Iron Bodhi"
1021                 ],
                  "Goal": "Pursuing wealth and power",
1022                 "Current Actions": "Leading a group to capture
1023                     ↪ Miao Qiaowei",
1024                 "Swordsmanship": "Tianlong Sect One-Stroke
1025                     ↪ Sword Technique",
                  "Condition": [
```

```
                        "Seriously injured",
                        "Bleeding profusely from the chest, in a
                            ↪ sorry state"
                ],
                "Sect": "Northern Branch of Tianlong Sect",
                "Aura": "Impressive",
                "Clothing": "Long robe and mandarin jacket",
                "Followers Count": "Eight",
                "Skills": [
                        "Swordplay",
                        "Pressure point striking"
                ],
                "Preferred Weapon": "Sword",
                "Grudge with Hu Yitong": "Had his treasure
                        ↪ blade taken and was struck to the ground
                        ↪ spitting blood"
            },
            "Relationships": {
                "Targets of Teasing": [
                        "Ma Chunhua",
                        "Xu Zheng"
                ],
                "Old Acquaintance": "Yan Ji",
                "Opponent": "Ma Xingkong",
                "Object of Fear": "Golden-faced Hero Miao",
                "Relationship with Miao Qiaowei": [
                        "Conflict",
                        "Rival, due to abducting Miao Qiaoweis
                            ↪ wife Nanlan"
                ],
                "Relationship with Nanlan": "Eloped",
                "Feelings Toward Nanlan": "Initially passionate
                        ↪ and infatuated, later diminished due to
                        ↪ her disdain",
                "Enemies": [
                        "Miao Qiaowei",
                        "Hu Yitong"
                ],
                "Rivals": [
                        "Hu Yitong",
                        "Li Tingbao"
                ],
                "Subordinates": [
                        "Warriors"
                ],
                "Daughter": "Tian Qingwen",
                "Comparison Target": "Hu Yitong",
                "Brother": "Tang Pei (Elder Brother)",
                "Challenger": "Tong Huaidao",
                "Elder Brother": "Tang Pei",
                "Chief Disciple": "Cao Yunqi",
                "Relationship with Fu Qilong": "Treated
                        ↪ respectfully by Fu Qilong",
                "Relationship with Tang Pei of Ganlin and Seven
                        ↪ Provinces": "Very close",
                "Relationship with Mr. Shi": "Acquainted and
                        ↪ have communicated",
                "Blinded Miao Qiaowei with Poison Grass": "Yes",
                "Searching for": "Nanlan"
            }
        }
```

## A.5.2 End-to-End Workflow Demonstration

Table 9: Example of the Question-Answer Process in EntiGraph and LoopAgent

| | |
|---|---|
| **EntiGraph** | |
| Original Context | Pierre Curie was a renowned French physicist known for his work on radioactivity. He was married to Marie Curie, who was also a distinguished scientist. Together, they conducted groundbreaking research that significantly advanced the understanding of radioactive elements. Pierre was born in Paris, France, where he also spent most of his life working and living. In 1903, Marie Curie was awarded the Nobel Prize in Physics, an achievement that highlighted their collaborative efforts in the field of radioactivity. Pierre himself made significant contributions to the study of magnetism and radioactivity... Marie Curie, born in Warsaw, Poland, moved to Paris to pursue her scientific career. She began her research on radioactivity, which led to the discovery of polonium and radium. In 1911, she was awarded the Nobel Prize in Chemistry for her services to the advancement of chemistry by the discovery of the elements radium and polonium, by the isolation of radium, and the study of the nature and compounds of this remarkable element... |
| Extracted Entities | **Entity**: Pierre Curie
Relationships:
    Wife: Marie Curie
Attributes:
    Occupation: Physicist
    Place of Birth: Paris, France
    Current Residence: Paris, France
    Work Location: Paris, France
Events:
    Event 1: 1903 - Wife Marie Curie awarded the Nobel Prize in Physics
    Event 2: Conducted significant research on radioactivity and magnetism
**Entity**: Marie Curie
Events:
    Event 1: Began research on radioactivity, discovered polonium and radium
    Event 2: 1911 - Awarded the Nobel Prize in Chemistry |
| Generated QA pairs (Micro) | Q: Who is Pierre Curie's wife?
A: Marie Curie. |
| Generated QA pairs (Feature) | Q: What were Pierre Curie's achievements?
A: Pierre Curie made important achievements in radioactivity and magnetism research, and he and his wife Marie Curie jointly studied radioactivity.
Q: What were Marie Curie's achievements?
A: Marie Curie made breakthrough progress in radioactivity research, discovered polonium and radium, and won the Nobel Prize in Physics and the Nobel Prize in Chemistry for this. |
| Generated QA pairs (Macro) | Q: Who won the Nobel Prize in Physics?
A: Marie Curie. |
| **LoopAgent** | |
| Query | What did Marie Curie and Pierre Curie contribute to science? |
| Sub-queries | - What did Marie Curie contribute to science?
- What did Pierre Curie contribute to science? |
| Sub-query Answers | - Marie Curie made breakthrough progress in radioactivity research, discovered polonium and radium, and won the Nobel Prize in Physics and the Nobel Prize in Chemistry for this.
- Pierre Curie made important achievements in radioactivity and magnetism research, and he and his wife Marie Curie jointly studied radioactivity. |
| If Answer is Incomplete | Generate new sub-queries based on related entities or events (not needed here). |
| Final Answer | Marie Curie conducted pioneering research on radioactivity, discovered polonium and radium, and won Nobel Prizes in Physics and Chemistry. Pierre Curie was a notable physicist who conducted research on radioactivity and magnetism, and contributed significantly to the understanding of radioactive elements. |

