# OpenReview forum: "Multi-Grained Knowledge for Retrieval-Augmented Question Answering on Hyper-long Contexts"
_ICLR.cc/2025/Conference — Submitted to ICLR 2025_

### Official Review · Reviewer_ad5h · 2024-11-01

**Soundness:** 2
**Presentation:** 3
**Contribution:** 2
**Rating:** 3
**Confidence:** 5

**Summary:**

This paper addresses a challenge in hyper-long context QA, where retrieving precise answers from extensive and dispersed content poses substantial obstacles. Current approaches often suffer from limitations, such as input-length constraints in LLMs and semantic loss in RAG systems. The authors propose a multi-grained entity graph-based QA framework, termed MKRAG, that operates across three granularity levels—micro, feature, and macro—to improve information extraction and reasoning across hyper-long texts. Their framework also introduces an iterative retrieval mechanism, LoopAgent, designed to refine retrievals and improve accuracy through multiple rounds. The evaluations across datasets show that MKRAG achieves state-of-the-art results, particularly excelling in scenarios with high granularity requirements, such as long-tail or detail-oriented queries.

**Strengths:**

1. The multi-grained QA method integrates a graph-based representation of entities and iterative retrieval, allowing the model to retain contextual coherence and retrieve relevant answers from hyper-long contexts.
2. The inclusion of LoopAgent, an iterative retrieval mechanism, improves QA accuracy by refining the search process across multiple rounds, which is particularly beneficial for complex and nuanced queries.
3. The model shows state-of-the-art performance on both hyper-long and moderately long datasets, outperforming baseline models like GPT-4 and other RAG-based approaches on LongBench and InfiniteBench.
4. The model has been effectively deployed in real-world applications, such as online novel-based platforms, and demonstrates enhanced scalability and practical utility in handling real-time queries.

**Weaknesses:**

1. Although the model circumvents some LLM limitations, the use of a multi-granularity framework and iterative retrieval adds complexity and computational demands, which may be prohibitive for broader real-time applications. How to evaluate the efficiency of the proposed method?
2. The approach heavily relies on accurate entity extraction and structured graph relationships. In cases where entity relationships are sparse or ambiguous, the model's performance may degrade. Do the authors test other entity extraction methods other than EntiGraph?
3. The paper should further discuss trade-offs between different granular levels and how the system decides the optimal level of granularity in real time, especially in cases with sparse information.
4. There are many prompt/context compressing methods that should be included as baselines:
4.1. Jiang, Huiqiang, et al. "Llmlingua: Compressing prompts for accelerated inference of large language models." arXiv preprint arXiv:2310.05736 (2023).
4.2. Jiang, Huiqiang, et al. "Longllmlingua: Accelerating and enhancing llms in long context scenarios via prompt compression." arXiv preprint arXiv:2310.06839 (2023).
4.3. Pan, Zhuoshi, et al. "Llmlingua-2: Data distillation for efficient and faithful task-agnostic prompt compression." arXiv preprint arXiv:2403.12968 (2024).
4.4. Li, Yucheng, et al. "Compressing context to enhance inference efficiency of large language models." arXiv preprint arXiv:2310.06201 (2023).

**Questions:**

See above

---

> ### Author Response · Authors · 2024-11-26
> **Responses to Reviewer ad5h**
>
> Dear reviewer ad5h,
>
> Thank you for taking the time to read our paper. We hope our responses adequately addresses your concerns.
>
> ---
> **Weakness1 - Efficiency of Our Method**
>
> Following your great advice, we evaluate the computational efficiency of our proposed method on InfiniteBench (a long-document dataset) and MultiFieldQA-zh (a relatively short-document dataset), and the results are shown as follows.
> | **Dataset**               | **Methods**        | **Time(s)** | **Accuracy (%)** | **F1 Score (%)** | **Avg. Iteration Rounds** |
> |---------------------------|--------------------|-------------|------------------|------------------|---------------------------|
> | **InfiniteBench (Long)**  | GPT-3.5 Turbo-16k  | 2.57        | 12               | 6.2              | 1                         |
> |                           | RAG (Ernie-3.5-8k) | 1.32        | 48               | 34.3             | 1                         |
> |                           | Ours               | 2.03        | 53               | 45.6             | 1.34                      |
> | **MultiFieldQA-zh (Short)**| GPT-3.5 Turbo-16k  | 1.91        | 74               | 61.2             | 1                         |
> |                           | RAG (Ernie-3.5-8k) | 1.80        | 75               | 60.7             | 1                         |
> |                           | Ours               | 1.89        | 78               | 65.6             | 1.09                      |
>
>
> Our method achieves significant improvements in accuracy and F1 scores compared to GPT-3.5Turbo-16k and RAG (Ernie-3.5-8k), while maintaining competitive inference times across both datasets. The average iteration rounds demonstrate the adaptability of our iterative retrieval mechanism, dynamically refining retrieval results based on query complexity. This configuration effectively balances retrieval quality and computational efficiency, showcasing the practicality of our approach for both long and short text scenarios.
>
> **Weakness2 - Other Entity Extraction Methods**
> Thank you for your insightful feedback. Before adopting EntiGraph, we tried other entity extraction approaches, such as BERT-based Named Entity Recognition (NER) models and various Graph Convolutional Network (GCN) methods. However, we found that these methods are limited to processing hyper-long contexts and complex relationships. We compare our EntiGraph with DeepKE, a representative method. The experimental results are shown as follows, which demonstrate the strong performance of our EntiGraph.
> | **Methods** | **Accuracy (%)** | **Recall (%)** | **F1 Score (%)** | **Extracted Entities (#)** |
> |-------------|-------------------|----------------|------------------|----------------------------|
> | DeepKE      | 52.9             | 18             | 27.4            | 164                        |
> | Ours        | 97.4             | 82             | 89.3            | 1,018                      |
>
>
> Although our experimental results demonstrate that EntiGraph ensures high precision and significantly improves recall, effectively addressing some limitations of traditional methods, we acknowledge that our work does not specifically focus on scenarios involving sparse or ambiguous entity relationships. We believe this represents an important direction for future research, and we plan to further investigate methods to better handle such cases and assess their impact on downstream tasks.

---

> ### Author Response · Authors · 2024-11-26
> **Responses to Reviewer ad5h**
>
> **Weakness3 - Different Granular Levels**
> Thanks for your suggestion, we have conducted additional ablation experiments focusing on micro, feature, and macro granularities across datasets.
>
> |     Dataset     | Granularity | Acc  (%) | F1 (%) | MKRAG (Acc) | MKRAG (F1) | $\Delta$ (Acc) | $\Delta$ (F1) |
> |:---------------:|:-----------:|:--------:|:------:|:-----------:|:----------:|:----------------:|:---------------:|
> |     DuReader    |    micro    |    63    |  22.4  |      73     |    31.4    |        +10       |       +9.0      |
> |                 |   feature   |    51    |  21.8  |             |            |        +22       |       +9.6      |
> |                 |    macro    |    64    |  23.7  |             |            |        +9        |       +7.7      |
> | MultiFieldQA-en |    micro    |    63    |  35.5  |      79     |    63.3    |        +16       |      +27.8      |
> |                 |   feature   |    65    |  33.0  |             |            |        +14       |      +30.3      |
> |                 |    macro    |    73    |  34.8  |             |            |        +6        |      +28.5      |
> | MultiFieldQA-zh |    micro    |    63    |  47.5  |      78     |    65.6    |        +12       |      +18.1      |
> |                 |   feature   |    59    |  44.6  |             |            |        +16       |      +21.0      |
> |                 |    macro    |    74    |  52.3  |             |            |        +1        |      +13.3      |
> |  InfiniteBench  |    micro    |   46.88  |  40.00 |      53     |    45.61   |       +6.12      |      +5.61      |
> |                 |   feature   |    42    |  34.70 |             |            |        +11       |      +10.91     |
> |                 |    macro    |    37    |  32.00 |             |            |        +16       |      +13.61     |
>
> These results illustrate the necessity of multi-grained representation for handling hyper-long contexts.
>
> **Weakness4 - Differences from Prompt/Context Compressing Methods**
> Thank you for your thoughtful suggestions and for highlighting relevant works on context compression. While the overarching goal of handling long texts aligns across both our method and the suggested compression-based approaches, the methodologies differ significantly. The mentioned methods primarily focus on compressing long contexts (typically around 1M tokens) to fit within the input constraints of existing models. In contrast, our approach is designed to directly address hyper-long contexts (e.g., InfiniteBench’s 2,068.6k tokens) by constructing and leveraging multi-grained entity graphs and iterative retrieval mechanisms. Regarding baseline selection, we prioritized algorithms with publicly reported metrics on hyper-long context benchmarks such as LongBench and InfiniteBench. As the suggested compression-based methods have not been evaluated on these datasets, they were not included in our comparisons. We deeply appreciate your recommendations and will explore integrating context compression techniques with our framework in future work to expand its applicability and performance across diverse scenarios.

---

### Official Review · Reviewer_qk3D · 2024-11-02

**Soundness:** 2
**Presentation:** 2
**Contribution:** 2
**Rating:** 6
**Confidence:** 3

**Summary:**

In this paper, a model called MKRAG is proposed to deal with hyper-long context QA tasks. Through multi-grained knowledge generation and iterative retrieval agent, MKRAG model can effectively extract and integrate information,  and thus improves the accuracy of question answering. The experimental results show that the MKRAG model achieves excellent performance on multiple benchmark datasets, and shows a strong ability of long text understanding and multi-domain question answering.

**Strengths:**

1.The MKRAG model captures the local and global information in the text by constructing multi-grained entity graphs (including micro, feature and macro levels), and can generate more accurate answers than traditional RAG methods.
2. Overall well written and easy to understand.
3.Iterative retrieval agent, MKRAG model uses LoopAgent iterative retrieval mechanism to refine the query through multiple rounds of retrieval to alleviate the problem of information fragmentation encountered by traditional methods in dealing with ultra-long text.
4.In this paper, based on the hyper-long context question and answer field, the experiments not only aim at the long text, but also verify that the proposed model is also applicable in the short text.

**Weaknesses:**

1.Authors have not clearly stated the key innovations of this paper.  Authors to explicitly state their key innovations and provide a clear comparison to existing methods, highlighting specific novel aspects of their approach, e.g., a specific new method or a new framework of well incorporating existing techniques.
2. The multi-grained entity graph and iterative retrieval, while effective, could be computationally intensive, limiting scalability in resource-constrained environments. Therefore, authors should conduct time and space complexity analyses, as well as perform corresponding experiments for verification, such as the Inference Time and  the Time per Iteration.
3.Some experimental details are missing, such as hardware specifications, number of iterations, the stopping condition for iterative retrieval, and the convergence criteria for all models.
4.  About " we employ context aggregation algorithms to integrate both local and global contexts of the same entity, and utilize an LLM, EntiGraph, to generate multi-grained QA pairs (i.e., micro-level, feature-level, and macro-level). This multi-level strategy not only ensures that the model produces highly accurate answers for complex queries, but also mitigates the fragmentation of information that often hampers traditional methods",  1) the "EntiGraph" is a LLM model? 2) why generate multi-grained QA pairs from micro-level, feature-level, and macro-level, and what are the insights about these? 3) why this multi-level strategy can ensure high accurate answers? 4) does this strategy lead to much noise?
5. The model is relatively complex, and the paper fails to clearly present the training and optimization process of the model. 1）How to perform joint optimization among multiple modules, i.e., ITERATIVE RETRIEVAL AGENT, MULTI-GRAINED REPRESENTATION, ENTITY EXTRACTION? How is the training data for each module constructed?  For each entity, how to construct the embeddings of its attributes, relationships, events and temporal information?

**Questions:**

1.Authors have not clearly stated the key innovations of this paper.  Authors to explicitly state their key innovations and provide a clear comparison to existing methods, highlighting specific novel aspects of their approach, e.g., a specific new method or a new framework of well incorporating existing techniques.
2. The multi-grained entity graph and iterative retrieval, while effective, could be computationally intensive, limiting scalability in resource-constrained environments. Therefore, authors should conduct time and space complexity analyses, as well as perform corresponding experiments for verification, such as the Inference Time and  the Time per Iteration.
3.Some experimental details are missing, such as hardware specifications, number of iterations, the stopping condition for iterative retrieval, and the convergence criteria for all models.
4.  About " we employ context aggregation algorithms to integrate both local and global contexts of the same entity, and utilize an LLM, EntiGraph, to generate multi-grained QA pairs (i.e., micro-level, feature-level, and macro-level). This multi-level strategy not only ensures that the model produces highly accurate answers for complex queries, but also mitigates the fragmentation of information that often hampers traditional methods",  1) the "EntiGraph" is a LLM model? 2) why generate multi-grained QA pairs from micro-level, feature-level, and macro-level, and what are the insights about these? 3) why this multi-level strategy can ensure high accurate answers? 4) does this strategy lead to much noise?
5. The model is relatively complex, and the paper fails to clearly present the training and optimization process of the model. 1）How to perform joint optimization among multiple modules, i.e., ITERATIVE RETRIEVAL AGENT, MULTI-GRAINED REPRESENTATION, ENTITY EXTRACTION? How is the training data for each module constructed?  For each entity, how to construct the embeddings for its attributes, relationships, events and temporal information?

---

> ### Author Response · Authors · 2024-11-26
> **Responses to Reviewer qk3D**
>
> Dear reviewer qk3D,
> Thank you for valuable suggestions! Please see the following for our point-by-point reply.
>
> ---
> **Weakness1 - The Key Innovations**
> Entity-Centric Multi-Grained Knowledge Representation: Traditional RAG methods, such as chunk-based retrieval, struggle to integrate fragmented semantic information and often lose global context. MKRAG addresses this by leveraging entity-centric multi-grained graph representations (micro-, feature-, and macro-level) to connect and synthesize both local and global contexts. This ensures that all relevant information is cohesively linked, overcoming input-length constraints and achieving precise, detailed answers for hyper-long context QA.
>
> Iterative Retrieval with Entity-Driven Reasoning: Single-pass retrieval methods lack the depth to resolve complex multi-entity questions, often leading to incomplete or incoherent results. MKRAG’s LoopAgent dynamically refines queries across multiple iterations by focusing on entities and their relationships. This iterative mechanism, combined with reasoning, ensures that the retrieval process captures all critical connections and provides semantically coherent results, significantly outperforming single-round or static retrieval strategies.
>
> Comprehensive Understanding for Contextual Coherence: Unlike shallow graph-based RAG approaches that focus on limited entity relationships, MKRAG uses its EntiGraph system to extract fine-grained entities, attributes, relationships, and events, linking them into a unified temporal and contextual framework. By pruning irrelevant entities and aggregating temporal information, MKRAG connects all dispersed details into a coherent structure. This holistic approach ensures that even hyper-long texts with scattered information are fully captured, providing detailed and contextually rich QA responses.
>
> **Weakness2 - The Scalability in Resource-Constrained Environments**
> Entity Extraction with Resource-Efficient "Offline" Processing: The entity graph and multi-grained representations are extracted in a manner that can be viewed as "offline". The one-time process significantly reduces runtime computational costs by amortizing the fixed extraction effort across multiple queries and users. This ensures scalability and efficiency in handling large datasets.
>
> Iterative Retrieval for Rare Complex Queries: The iterative retrieval mechanism is primarily designed to address rare and highly complex queries through query rewriting and refinement. For experiment datasets, the average number of retrieval iterations remains low (e.g., 1.34 for InfiniteBench and 1.09 for MultiFieldQA-zh), meaning the approach adds negligible overhead while maintaining the flexibility to adapt to outlier cases.
>
> **Weakness3 - Experimental Details**
> Hardware specifications: Our experiments were conducted on a cluster with Nvidia A100 GPUs (40GB, 4 cards).
> Stopping condition for iterative retrieval: The average number of iterations in the iterative retrieval process is 1.34 on InfiniteBench and 1.09 on MultiFieldQA-zh, as shown in Table 6 of the paper. The stopping condition is determined by the LoopAgent, which evaluates whether the retrieved information is related to the query. If relevant, the system outputs the answer; otherwise, it rewrites the query for further refinement.
> Convergence Criteria: Since only two modules, Entity Extraction and Multi-Grained Representation, undergo fine-tuning, their convergence is determined by the performance on validation set.

---

> ### Author Response · Authors · 2024-11-26
> **Responses to Reviewer qk3D**
>
> **Weakness4 - Model Details**
> 1) EntiGraph: Yes, EntiGraph is a distilled lightweight LLM, specifically optimized for entity extraction and multi-grained representation tasks.
> 2) Multi-grained QA pairs: The purpose of generating multi-grained QA pairs is to comprehensively extract information from the text. At the micro-level, the goal is to capture fine-grained details, such as specific attributes or relationships for entities. The feature-level expands this by integrating more attributes, relationships, and events to provide contextual richness. The macro-level combines all this information globally, enabling reasoning across entities. For instance, the model can leverage relationships like A is related to B and A is related to C to infer connections between B and C, thus creating a holistic view of the information.
> 3) The strength of multi-grained strategy: This strategy ensures highly accurate answers by addressing the fragmentation of information often seen in long and complex texts. By dynamically combining local and global contexts, the method connects dispersed information, reducing loss and enhancing reasoning capabilities. Additionally, the multi-level granularity supports adaptability to various query types, from detail-specific to complex reasoning.
> 4) The redundancy of multi-grained strategy: While this strategy may introduce some redundancy, it is managed through mechanisms like entity pruning (Section 4.3) and adaptive granularity selection, which minimize unnecessary outputs. These measures ensure that only relevant QA pairs are used during retrieval, balancing comprehensiveness and efficiency.
>
>
> **Weakness5 - The Optimization Process of The Model**
> Joint Optimization of Modules:The model is designed to allow modular operation without requiring joint optimization. This design ensures flexibility and adaptability to various production scenarios while reducing overall complexity.
> The Entity Extraction Module and Multi-Grained Representation Module are optimized independently through fine-tuning and aligned data processing.
> The Iterative Retrieval Agent dynamically refines queries and integrates results without requiring training, leveraging the representations from the other modules. Training Data Construction in entity extraction:
>   a. Coarse-Labeled Data Preparation: We used GPT-4 to extract entities from texts in domains such as novels and financial reports, focusing on two categories: key entities and generic entities.
>   b. Manual Refinement: The coarse-labeled data was manually screened and annotated, resulting in a high-quality dataset of approximately 3k samples.
>   c. Synthetic Sample Generation: GPT-4 was further used to generate additional samples by synthesizing attributes, relationships, and events for annotated entities.
> Multi-Grained Representation Module: Data from the entity extraction process was used to construct QA pairs across micro, feature, and macro granularity levels, ensuring alignment with the downstream question-answering task.

---

### Official Review · Reviewer_jrNj · 2024-11-03

**Soundness:** 1
**Presentation:** 3
**Contribution:** 2
**Rating:** 5
**Confidence:** 4

**Summary:**

To address the challenges of hyper-long context QA, particularly the limitations of context windows and retrieval errors in retrieval-augmented generation (RAG) due to inadequate semantic representation, this paper proposes Multi-grained Knowledge RAG Method (MKRAG). MKRAG involves extracting entities from context with EntiGraph chunk-by-chunk, generating multi-grained QA pairs, and iteratively retrieve related information by refining queries to get the final answer.

Results show that the proposed method achieve on par or slightly better performance compared with SOTA models/methods, and achieves high performance gain in the online test.

**Strengths:**

1. The EntiGraph module effectively enhances the representation and may have the potential to extract most necessary information from the document.
2. The multi-grained knowledge generation from entity graph helps capture possible QA from different granularity.

**Weaknesses:**

1.	This paper claims that the EntiGraph module enables accessing nearly all the necessary information. However, the performance of EntiGraph is not discussed.
2.	In the Entity Pruning section, how the threshold $\tau$ is selected is not explained. There are also pre-defined thresholds in (10). The selection of thresholds are not justified by detailed analysis of module performance.
3.	By turning the document into entity graphs and generating knowledge accordingly, the model risks missing information during these processes, and the agent may be unable to answer the question given the insufficient information.
4.	In the ablation study section, “Chuck Retrieval + Baseline” setting uses a chunk size of 500 tokens/chunk while the context window limit is 4k. It is not clear what is the chunk size used in MKRAG and if the small chunk size in the previous setting limits its performance.
5.	This paper claims that the proposed method demonstrating high accuracy in Online Test and high user satisfaction rate. However, details of the tests are not provided and the results are not compared against other SOTA models/methods.

**Questions:**

1.	How is "importance weight" defined in equation (6)? More explanation of this part would be helpful.
2.	What is the module performance in each subtask? e.g., the recall rate of entities, accuracy of entity aggregation, effectiveness of entity pruning, effectiveness of thresholds in equation (10), and more?
3.	What is the average number of rewrites for each question (line 401)? How is the computation cost for the LoopAgent?
4.	The hyter-long context QA benchmark, InfiniteBench, has an average output of 6.3 tokens, which should be only a few words. Does these questions really need query-decomposition described in line 383?

---

> ### Author Response · Authors · 2024-11-26
> **Responses to Reviewer jrNj**
>
> Dear reviewer jrNj,
>
> Thank you so much for these very valuable and constructive suggestions! Please kindly find the point-to-point responses below.
>
> ---
> **Weakness1 - Effectiveness of Information Extraction Module & Weakness3 - Information Missing of EntiGraph**
>
> Thanks for your feedback!
>
> The performance of the EntiGraph module is thoroughly evaluated in the revised manuscript, as shown below:
>
> | **Methods** | **Accuracy (%)** | **Recall (%)** | **F1 Score (%)** | **Extracted Entities (#)** |
> |-------------|-------------------|----------------|------------------|----------------------------|
> | DeepKE      | 52.9             | 18             | 27.4            | 164                        |
> | Ours        | 97.4             | 82             | 89.3            | 1,018                      |
>
> Compared to BERT-based methods, EntiGraph achieves a significantly higher F1 score of 89.3% (vs. 27.4%) and extracts six times more entities. This ensures comprehensive coverage and reliability of the information used for downstream tasks. Information loss is addressed by leveraging a specialized entity extraction model to maximize the capture of necessary details.
>
>
>
> **Weakness2 - Effect of threshold $\tau$**
> We conduct experiments to analyze the effect of threshold $\tau$, and the results are as follows.
>
>
> | **Thresholds (τ)** | **Precision (%)** | **Recall (%)** | **F1 (%)** | **nums (Entities Retained)** |
> |---------------------|-------------------|----------------|------------|-----------------------------|
> | 0                  | 86.3             | 84.6           | 85.5       | 1272                        |
> | 0.3                | 90.5             | 80.8           | 85.3       | 1107                        |
> | **0.5**            | **95.2**         | **80.8**       | **87.5**   | **688**                     |
> | 0.7                | 88.2             | 65.4           | 75.1       | 398                         |
> | 1                  | 90.1             | 42.3           | 57.7       | 268                         |
>
>
> **Weakness4 - Chuck Size**
> The chunk size used in MKRAG is 500 tokens. We recall top-k chunks, not only one. Hence, the chunk size cannot be set to 4k tokens. Besides, the longer the chunk size, the more scattered the information the model needs to pay attention to. After experiments, we found that the chunk size of 500 tokens is the most appropriate.
>
>
> **Weakness5 - Online Test**
> We have reported the result of online test in Section 5.4. However, due to data sensitivity, it is recommended not to evaluate other baseline models on the online platform. Hence, we conduct experiments on public datasets and compare the experimental results with them to illustrate the effectiveness of our approach.

---

> ### Author Response · Authors · 2024-11-26
> **Responses to Reviewer jrNj**
>
> **Question1 - The Weights of Entities**
> $w_k$ represents the importance weight of each entity. We define four weight types as inputs for the baseline LLM (Ernie-3.5-8k): general or ambiguous entities (e.g., "villager" or "merchant") - 0.3; related to the context, describing a specific entity (e.g., "Zhu Bajie's wife") - 0.5; rare or unique entities, often with unique names or backgrounds (e.g., "White Bone Demon") - 0.7; core entities, including only entities that are indispensable to the context (e.g., "Sun Wukong") - 1. Thanks for your advice, we will add this part in the manuscript.
>
> **Question2 - Module Performance in Subtask**
> We evaluated the performance of each module, and the results are shown below. All details have been added to the revised manuscript.
> ### Entity Extraction Module
>
> | **Methods** | **Accuracy (%)** | **Recall (%)** | **F1 Score (%)** | **Extracted Entities (#)** |
> |-------------|-------------------|----------------|------------------|----------------------------|
> | DeepKE      | 52.9             | 18             | 27.4            | 164                        |
> | Ours        | 97.4             | 82             | 89.3            | 1,018                      |
>
> ### Effectiveness of Entity Pruning
> The experiment demonstrates that a pruning threshold (τ) of 0.5 achieves the best balance between Precision and Recall, detailed in Appendix.
>
> | **Thresholds (τ)** | **Precision (%)** | **Recall (%)** | **F1 (%)** | **nums (Entities Retained)** |
> |---------------------|-------------------|----------------|------------|-----------------------------|
> | 0                  | 86.3             | 84.6           | 85.5       | 1272                        |
> | 0.3                | 90.5             | 80.8           | 85.3       | 1107                        |
> | **0.5**            | **95.2**         | **80.8**       | **87.5**   | **688**                     |
> | 0.7                | 88.2             | 65.4           | 75.1       | 398                         |
> | 1                  | 90.1             | 42.3           | 57.7       | 268                         |
> ### Performance of Multi-Grained Representation
> | **Dataset**                 | **Granularity** | **Acc (%)** | **F1 (%)** | **MKRAG (Acc)** | **MKRAG (F1)** | **$\Delta$ (Acc)** | **$\Delta$ (F1)** |
> |-----------------------------|-----------------|-------------|------------|-----------------|----------------|----------------------|---------------------|
> | **DuReader**                | micro           | 63          | 22.4       | 73              | 31.4           | +10                  | +9.0                |
> |                             | feature         | 51          | 21.8       |                 |                | +22                  | +9.6                |
> |                             | macro           | 64          | 23.7       |                 |                | +9                   | +7.7                |
> | **MultiFieldQA-en**         | micro           | 63          | 35.5       | 79              | 63.3           | +16                  | +27.8               |
> |                             | feature         | 65          | 33.0       |                 |                | +14                  | +30.3               |
> |                             | macro           | 73          | 34.8       |                 |                | +6                   | +28.5               |
> | **MultiFieldQA-zh**         | micro           | 63          | 47.5       | 78              | 65.6           | +12                  | +18.1               |
> |                             | feature         | 59          | 44.6       |                 |                | +16                  | +21.0               |
> |                             | macro           | 74          | 52.3       |                 |                | +1                   | +13.3               |
> | **InfiniteBench**           | micro           | 46.88       | 40.00      | 53              | 45.61          | +6.12                | +5.61               |
> |                             | feature         | 42          | 34.70      |                 |                | +11                  | +10.91              |
> |                             | macro           | 37          | 32.00      |                 |                | +16                  | +13.61              |

---

> ### Author Response · Authors · 2024-11-26
> **Responses to Reviewer jrNj**
>
> **Question3 - Average Number of Rewrites &  Computation Cost of LoopAgent**
> We supplemented experiments and the results have been added to the revised manuscript. The details are shown below:
> | **Dataset**               | **Methods**        | **Time(s)** | **Accuracy (%)** | **F1 Score (%)** | **Avg. Iteration Rounds** |
> |---------------------------|--------------------|-------------|------------------|------------------|---------------------------|
> | **InfiniteBench (Long)**  | GPT-3.5 Turbo-16k  | 2.57        | 12               | 6.2              | 1                         |
> |                           | RAG (Ernie-3.5-8k) | 1.32        | 48               | 34.3             | 1                         |
> |                           | Ours               | 2.03        | 53               | 45.6             | 1.34                      |
> | **MultiFieldQA-zh (Short)**| GPT-3.5 Turbo-16k  | 1.91        | 74               | 61.2             | 1                         |
> |                           | RAG (Ernie-3.5-8k) | 1.80        | 75               | 60.7             | 1                         |
> |                           | Ours               | 1.89        | 78               | 65.6             | 1.09                      |
>
> On the test datasets, the maximum iteration rounds were set to 2, with average iteration rounds of 1.34 (InfiniteBench) and 1.09 (MultiFieldQA-zh). The inference time for our method ranged from 1.89s to 2.03s.
>
> **Question4 - Short Tokens**
> We only decompose the query if it contains multiple entities. If the query contains only one entity, the number of subquery is 1.

---

### Official Review · Reviewer_vkcq · 2024-11-05

**Soundness:** 2
**Presentation:** 3
**Contribution:** 3
**Rating:** 3
**Confidence:** 3

**Summary:**

This paper introduces a multi-grained entity graph-based QA method that constructs an entity graph and dynamically combines both local and global contexts, capturing information across three granularity levels: micro, feature, and macro levels, and incorporates iterative retrieval and reasoning mechanisms to generate accurate answers for hyper-long contexts. Evaluation results on LongBench and InfiniteBench demonstrate the effectiveness of the approach, significantly outperforming existing methods in both the accuracy and granularity of the extracted answers, and it can be deployed in online novel-based applications.

**Strengths:**

1. This paper proposes MKRAG (Multi-grained Knowledge Retrieval-Augmented Generation) for hyper-long context question answering. By integrating multi-grained entity graphs with an iterative retrieval and reasoning process, MKRAG addresses the limitations of traditional models constrained by context window size.
2. This paper introduces LoopAgent, an iterative retrieval framework that progressively refines queries across multiple retrieval cycles. By incorporating advanced reasoning capabilities, LoopAgent improves both retrieval and answering accuracy and mitigates information loss in traditional single-pass retrieval methods, particularly in complex multiple-entity scenarios.

**Weaknesses:**

1. The entities and their associated attributes, relationships, and events are extracted by LLMs. However, as noted in previous work, LLMs may fall short in information extraction (IE) tasks, such as entity extraction, relation extraction, and event extraction. If LLMs cannot handle IE well, errors could propagate through the system, leading to random, unpredictable, and non-generalizable outcomes. It could be better if the authors provide evaluation on the reliability of the IE process.
2. While the model of experiments is based on ERNIE, there lack a comparison with the other ERNIE variants capable of handling longer inputs, such as ERNIE-Turbo-128K. Including a comparison with models from the same series would strengthen the paper by better demonstrating the effectiveness of the proposed approach.

**Questions:**

See "Weaknesses".

---

> ### Author Response · Authors · 2024-11-26
> **Responses to Reviewer vkcq**
>
> Dear reviewer vkcq,
> Thank you so much for these very insightful and constructive comments, please see the following for our point-by-point responses.
>
> ---
> **Effectiveness of Information Extraction Module**
>
> Following your great advice, we conduct more experiments to analyze the reliability of information extraction module, and the results are shown as follows.
> | **Methods** | **Accuracy (%)** | **Recall (%)** | **F1 Score (%)** | **Extracted Entities (#)** |
> |-------------|-------------------|----------------|------------------|----------------------------|
> | DeepKE      | 52.9             | 18             | 27.4            | 164                        |
> | Ours        | 97.4             | 82             | 89.3            | 1,018                      |
>
> Our experiments demonstrate that LLM-based methods significantly outperform BERT-based methods across all metrics. Additionally, the survey "Large Language Models for Generative Information Extraction: A Survey" (https://arxiv.org/pdf/2312.17617) highlights extensive work leveraging LLMs for IE tasks, including entity extraction, relation extraction, and event extraction. These studies and results collectively demonstrate the superiority and reliability of LLMs in information extraction tasks.
>
> **Comparison with other ERNIE variants**
>
> The variants of ERNIE do not contain ERNIE-Turbo-128K, andd the maximum length of ERNIE-Turbo is 8k. ERNIE has a simple version of ERNIE-Speed-128K. ERNIE-Speed-128K has longer input tokens and faster inference speed, but its performance is not better than the baseline model (i.e., Yi-6B-200K, Kimi-Chat, Chatglm3-6B-128K). We have compared our method with these baseline models, and the experimental results demonstrate the effectiveness of our model.

---

### Meta-Review · Area_Chair_4oWd · 2024-12-19

**Metareview:**

The paper proposes a novel method, Multi-Grained Knowledge Retrieval-Augmented Generation (MKRAG), aimed at improving question answering on hyper-long contexts, a challenging problem where large amounts of information need to be synthesized for accurate responses. MKRAG integrates a multi-grained entity graph (EntiGraph) for information extraction and employs an iterative retrieval mechanism (LoopAgent) to refine queries over multiple cycles, which helps enhance the retrieval precision and overall QA performance. The results show that MKRAG outperforms existing state-of-the-art methods on benchmark datasets and exhibits practical utility in online applications for handling long-tail queries and complex, multi-entity scenarios.

Strengths:

1. Innovative Approach: MKRAG offers a novel solution to overcoming the limitations of context window sizes in language models (LMs) through the use of a multi-grained entity graph for efficient information extraction and retrieval. This is an important step forward in improving the performance of QA systems on long-form documents.
2. Effective Iterative Retrieval: The LoopAgent mechanism, which refines queries iteratively, is a unique and valuable contribution. It improves retrieval accuracy and ensures that the model can answer complex questions that require precise extraction from large datasets.
3. Strong Experimental Results: The paper demonstrates robust empirical results on LongBench and InfiniteBench, showcasing the efficacy of MKRAG over state-of-the-art methods.
4. Real-World Applicability: The successful implementation of MKRAG in online applications further strengthens its practical value and potential impact.

Weaknesses:

1. Information Extraction Reliability: While the EntiGraph module is a key part of the model, the paper lacks a detailed analysis of the reliability of entity extraction, especially given that LLMs often struggle with tasks like entity and relation extraction. The reliability of this component needs to be more thoroughly evaluated to understand potential errors and limitations.
2. Limited Comparison with ERNIE Variants: While the paper uses ERNIE for experiments, it fails to compare the method with other ERNIE variants, particularly those designed for longer inputs (e.g., ERNIE-Turbo-128K). A comparison with these variants would have provided a more well-rounded understanding of MKRAG’s strengths.
3. Methodology Gaps:
   - The threshold selection for pruning entities is not sufficiently explained, leaving a gap in understanding its impact on performance.
   - The ablation study is underdeveloped, particularly regarding the chunk size of 500 tokens used in MKRAG. Justification of this choice in relation to context window limits would improve the methodology's transparency.
4. Lack of Evaluation Transparency: The online test results are mentioned, but the paper does not provide sufficient detail about the test setup, comparison with other models, or specific outcomes. This diminishes the credibility of the claims.
5. Missing Quantitative Analysis of Individual Components: While the model is evaluated overall, there is no in-depth analysis of the performance of individual components, such as entity extraction and aggregation. Including metrics like recall, accuracy, and precision for these modules would help better assess the method’s overall effectiveness.

While MKRAG introduces a promising and innovative approach to hyper-long context QA, there are weaknesses that hinder its overall presentation and robustness. The paper would benefit from greater clarity regarding key decisions, a more thorough evaluation of its components, and a deeper comparison with other methods.

**Additional Comments On Reviewer Discussion:**

1.Innovation and Contribution:
   Reviewers generally agree that MKRAG offers an innovative approach, especially through its multi-grained entity graph (EntiGraph) and iterative retrieval mechanism (LoopAgent). These components represent significant contributions to addressing the challenges posed by hyper-long context QA tasks. However, there was a consensus that the novelty of the approach needs to be more explicitly highlighted in the paper. While the approach is promising, its uniqueness relative to existing solutions (e.g., ERNIE variants and other long-context models) should be more clearly articulated.

2. Methodological Clarity:
   Several reviewers pointed out gaps in the methodological description. In particular, the selection of pruning thresholds and the justification for the chunk size of 500 tokens were questioned. The lack of detailed explanation for these choices weakens the methodology. Reviewers suggested that providing experimental justification for these decisions, perhaps through additional ablation studies, would help clarify their importance and impact on performance.

3. Information Extraction and Reliability:
   There was a common concern about the reliability of the information extraction process, especially with respect to LLM-based methods like EntiGraph. While the results are promising, there is a lack of analysis on how errors in entity and relation extraction might affect the model's overall accuracy. A more thorough exploration of potential failure modes and how MKRAG mitigates these would enhance the credibility of the approach. Reviewers suggest this could be addressed by providing more robust experiments or discussion on the reliability and robustness of the entity extraction component.

4. Comparative Evaluation:
   A key point of discussion among the reviewers was the limited comparison with other ERNIE variants and other state-of-the-art models. While MKRAG demonstrates strong results, a broader comparison to other long-context handling methods would provide a clearer picture of its relative strengths and weaknesses. Some reviewers suggested that the inclusion of comparisons with ERNIE-Turbo-128K or similar variants could add depth to the evaluation.

5. Experimental Transparency:
   The transparency of experimental setup, particularly with respect to online testing, was identified as a significant concern. Reviewers noted that the paper mentions successful online tests but does not provide sufficient details on the experimental conditions or how MKRAG compares to baseline models. Without these details, the claims about real-world applicability are less convincing. Reviewers emphasized the importance of providing specifics such as hardware configurations, iteration settings, and detailed performance metrics in online scenarios.

6. Model Complexity and Optimization:
   The complexity of MKRAG, including the integration of the multi-grained entity graph, retrieval mechanism, and reasoning modules, was highlighted as both a strength and a potential weakness. Some reviewers felt that the paper could benefit from a clearer discussion on how these components interact and are optimized. Specifically, the joint optimization of the different modules and the training data for each component need to be clarified. Providing insights into the model’s optimization process would help potential users better understand its practical challenges.

7. Quantitative Analysis of Modules:
   Finally, several reviewers suggested including a more detailed quantitative analysis of the performance of individual modules (such as entity extraction, aggregation, and pruning). While the overall performance of MKRAG is presented, the lack of breakdowns for individual components leaves uncertainty about which parts of the model contribute most to its success. Including this analysis could help pinpoint the model’s strengths and areas for further improvement.

The reviewers collectively recognize the promise of MKRAG in addressing the hyper-long context QA challenge. However, to strengthen the paper and make a more compelling case for its acceptance, the authors must address the weaknesses highlighted by the reviewers. This includes improving methodological clarity, expanding the experimental evaluation (particularly online tests), and providing more detailed analysis of the individual components' performance. With these revisions, MKRAG could represent a significant contribution to the field of retrieval-augmented question answering.

---

### Decision · Program_Chairs · 2025-01-22

Reject